# The Sentiment Problem: A Critical Survey towards Deconstructing Sentiment Analysis

**Pranav Narayanan Venkit**[1*]  **Mukund Srinath**[1*]  **Sanjana Gautam**[1]
**Saranya Venkatraman**[1]  **Vipul Gupta**[2]  **Rebecca J. Passonneau**[2]  **Shomir Wilson**[1]

[1] College of Information Sciences and Technology
[2] Department of Computer Science & Engineering, College of Engineering
Pennsylvania State University

{pranav.venkit, mukund, sanjana.gautam, saranyav, vkg5164, rjp49, shomir}@psu.edu

## Abstract

We conduct an inquiry into the sociotechnical aspects of sentiment analysis (SA) by critically examining 189 peer-reviewed papers on their applications, models, and datasets. Our investigation stems from the recognition that SA has become an integral component of diverse sociotechnical systems, exerting influence on both social and technical users. By delving into sociological and technological literature on sentiment, we unveil distinct conceptualizations of this term in domains such as finance, government, and medicine. Our study exposes a lack of explicit definitions and frameworks for characterizing sentiment, resulting in potential challenges and biases. To tackle this issue, we propose an ethics sheet encompassing critical inquiries to guide practitioners in ensuring equitable utilization of SA. Our findings underscore the significance of adopting an interdisciplinary approach to defining sentiment in SA and offer a pragmatic solution for its implementation.

## 1 Introduction

Sentiment Analysis (SA) has emerged as a significant research focus in Natural Language Processing (NLP) over the last decade. It has now become an indispensable tool in discerning opinions and emotions in written text (Medhat et al., 2014), evaluating social entities' reputation (Yuliyanti et al., 2017), analyzing and predicting financial needs (Wang et al., 2013), and aiding in effective political decision-making (Cardie et al., 2006). This is illustrated in Figure 1 which shows the rising numbers of peer-reviewed articles on sentiment analysis published in SCOPUS every year.

Existing research reveals a notable absence of interdisciplinary endeavors to comprehend the social dimensions of SA, encompassing aspects like emotion and fairness (Mohammad, 2022; Blodgett

* Authors Contributed Equally

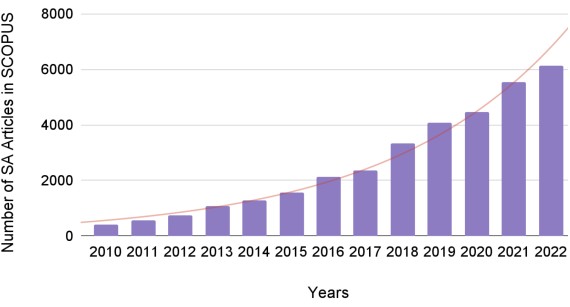

Figure 1: Number of articles published each year (from 2010 to 2022) in SCOPUS that contain the term 'sentiment analysis' in the title, abstract, or keywords.

et al., 2020). This lack of collaborative thinking has resulted in flawed analyses and biased outcomes. Given the extensive range of applications of SA spanning diverse domains such as healthcare, finance, and policymaking, it is crucial to avoid replicating such tendencies. Furthermore, SA, despite addressing social constructs like emotion, subjectivity, and opinion, has been limited in its incorporation of psychological and sociological definitions of sentiment (Stark and Hoey, 2021). While numerous studies have examined the utilization of SA, encompassing its inherent challenges and future directions (Cardie et al., 2006; Zhang et al., 2022), the interdisciplinary and sociotechnical dimensions of SA have received limited exploration.

To this end, we explore this gap in the literature by examining sentiment through a technical perspective concentrating on the evolution of SA into a social system. We then evaluate sentiment, examining the various definitions of sentiment through a sociotechnical lens. We also investigate the application of SA, presenting insights into its utilization. These investigations will shed light on the interdisciplinary divide of the term sentiment. Next, we evaluate the motivation behind establishing necessary frameworks for measuring sentiment by examining various different SA models and datasets.

Through our critical survey of 189 unique works

| Categories | Frequency |
|---|---|
| Sentiment Analysis Applications | 60 |
| Sentiment Analysis Models | 43 |
| Sentiment Analysis Datasets | 19 |
| Surveys and Meta-Analysis | 21 |
| Frameworks | 17 |
| Others | 29 |

Table 1: Frequency of papers reviewed for each category of the works in SA.

in SA (as shown in Table 1, we reveal that very few works (<5%) in SA try to explicitly define sentiment and sentiment analysis. Our results highlight a lack of effort within the field of NLP to understand the interdisciplinary aspect of sentiment. We also show an absence of synchronization in the field, leading to multiple variations of the term sentiment. Our analysis illustrates how such systems can cause sociodemographic biases due to the lack of nuance in measuring sentiment. To mitigate this issue of an interdisciplinary gap, we propose an ethics sheet (Mohammad, 2022) consisting of ten critical questions to be used as a metaphorical 'nutrition label' to understand the issues of SA models by both the user as well as the developers alike.

## 2 A Survey of Surveys

We now *chronologically analyzing various surveys* published in the field of NLP. Medhat et al. (2014) surveyed 54 articles and categorized them based on utility. They showed that SA was synonymous with opinion mining and subjective analysis, and was primarily utilized to analyze product reviews. Similarly, Alessia et al. (2015) presented a summary of SA, stating it to have evolved into a sociotechnical system (Prun and Raymond, 2021) often used in the fields of politics, public actions, and finance. Further, Ribeiro et al. (2016) reviewed *SA models* and benchmarked a comparison of 24 SA models. They found that most models were developed to measure sentiment in social posts, product reviews, and texts in news articles. However, the metrics of measurement varied considerably across datasets and models, highlighting the need for uniformity in the field of SA.

With the advent of deep learning, more SA models were developed using deep learning architectures, as summarized by Zhang et al. (2018). The work demonstrated how similar architectures could now be used in applications such as emo-

tion analysis, sarcasm analysis, and toxicity analysis. Sánchez-Rada and Iglesias (2019) surveyed the social context of sentiment analysis, reviewing its applications, limitations, and utilities as a sociotechnical system. Drus and Khalid (2019) surveyed works on SA from 2014 to 2019 to understand its social utility. They found that most of the work in SA was used in interdisciplinary contexts related to world events, healthcare, politics, and business.

Recent surveys by Birjali et al. (2021); Guo et al. (2021); Wankhade et al. (2022); Zad et al. (2021) provide up-to-date perspectives on SA reflecting a shift towards fine-grained approaches, including deep learning and aspect-based sentiment analysis, enabling a more contextual understanding of sentiment. Similarly, recent works by Zhang et al. (2022); Soni and Rambola (2022) have specifically focused on aspect-based sentiment analysis and implicit aspect detection methods. Overall, these surveys reflect a scoping of sentiment analysis to include *people's sentiments, opinions, attitudes, evaluations, appraisals, and emotions towards services, products, individuals, organizations, issues, topics, events, and their attributes*. However, none of these works discuss the interdisciplinary framework of emotion or sentiment.

## 3 Examination of Sentiment

We start by analyzing the various sentiment frameworks in SA and comparing them to existing social frameworks. By doing so, we aim to uncover the distinctions between the different notions of this term, shedding light on the gap between the technical and social aspects of sentiment. In this context, we define a sociotechnical system as a composite of social and technical components that collectively contribute to goal-oriented behavior, impacting both social and technical actors engaged with the system (Cooper and Foster, 1971). Throughout this work, we use the term 'framework' to denote a conceptual structure or set of principles that offer guidance for measuring or defining a specific concept within a study.

### 3.1 The Technical Perception of Sentiment

The phrase *sentiment analysis* likely originated from its first use case in NLP to analyze market sentiment (Das and Chen, 2001). The authors attempted to classify stock ratings based on opinions on a message board. Similarly, Turney and Littman

| Framework | Definitions | Example |
|-----------|-------------|---------|
| Semantic Orientation | Measure of whether the words or expressions used in a text convey a positive or negative meaning | (Agarwal et al., 2016) |
| Opinions or Evaluations | Author's attitude towards a topic | (Zhai et al., 2011) |
| Affect or Feeling | Author's disposition towards a specific theme | (Birjali et al., 2021) |
| 3-D polarity | Framework with 3 dimensions of polarities: Subjective\Objective, Positive\Negative, Strength | (Sebastiani and Esuli, 2006) |
| Emoticons | Emoticons as sentiment indicators | (Lou et al., 2020) |
| Object's orientation | Measure of the attitude towards individual aspects of an entity | (Mowlaei et al., 2020) |
| Implicit | Emotional tendencies implied by commonsense knowledge of the effect of concepts or events | (Zhang and Liu, 2011) |
| Human Annotation | Sentiment ratings collected from experts or crowd-sourced data collection | (Kenyon-Dean et al., 2018) |

Table 2: Frameworks of Sentiment and corresponding definitions in Sentiment Analysis

(2002) experimented with using the **semantic orientation** of words to find whether product reviews are positive or negative. Readily available data in the form of product reviews on e-commerce websites influenced early SA works and firmly established it to almost exclusively mean opinion mining, with sentiment defined as: *'overall opinion towards the subject matter'* (Pang et al., 2002).

Following this, Read (2005) proposed the use of **emoticons** as a proxy for ground truth data to measure sentiment in text. They defined SA as the method to *'identify a piece of text according to its author's general feeling toward their subject, be it positive or negative.'* This marked a stark deviation of SA from 'opinion mining.' This expansion of the meaning of sentiment can also be seen in the work of Wilson et al. (2005b) where they defined SA as *'the task of identifying positive and negative opinions, emotions, and evaluations'*. Subsequently, Sebastiani and Esuli (2006) proposed that SA consists of **three dimensions**: *subjective-objective polarity, positive-negative polarity,* and *strength of polarity*.

The first use of SA as a sociotechnical system is marked by Go et al. (2009)'s approach to train a SA model using data from a social media platform, namely Twitter. While most prior work still treated SA as a method to extract an author's subjective or objective opinion regarding an entity or an object, Go et al. (2009) defined sentiment from the perspective of a general **feeling or emotion** in text. Their definition of sentiment as *'a personal positive or negative feeling or opinion'*, is a marked deviation that influenced much of the literature in SA. Maas et al. (2011)'s work recognized sentiment as a 'complex, multi-dimensional concept' and attempted to operationalize it through a vector representation. Similarly, Zhang and Liu (2011) defined sentiment as an *'emotional tendency implied by commonsense knowledge of the effect of concepts or events'* to define an implicit form of sentiment. To quantify sentiment from a 'human perspective', Kenyon-Dean et al. (2018) used **human annotation**, as a methodology to define and measure sentiment, using crowd-sourced data.

Table 2 tabulates the multifarious frameworks encountered in SA. Here we see that SA does not follow a well-defined comprehensive framework. With the evolution of the field, different researchers adapted SA in dissimilar ways while not making a clear distinction between concepts such as emotions, opinions, and attitudes. We posit that there is a need for a nuanced, socially informed, and theoretically motivated framework for sentiment in SA. To understand sentiment from an interdisciplinary perspective and draw out an interdisciplinary framework, we examine its meaning from a sociological perspective.

## 3.2 The Social Perception of Sentiment

A notable distinction exists between computational and psycho-linguistic perspectives on sentiment. In psychology, sentiment is often defined as *"socially constructed patterns of sensations, expressive gestures, and cultural meanings organized around a relationship to a social object, usually another person or group such as a family."* (Gordon, 1981). While sentiment is most commonly categorized as positive, negative, or neutral in computational literature, it encompasses a broader spectrum, ranging from mild to intense (Taboada, 2016; Jo et al., 2017). Furthermore, sentiment (in psychology) is captured through physiological indicators, like facial expressions and heart rate variability (Wiebe et al., 2005; Plutchik, 2001).

Psychological research widely recognizes that a simplistic positive-negative dichotomy is *inade-*

*quate* for capturing the intricate range of human emotions (Hoffmann, 2018). This is evident in the distinction between seemingly negative emotions such as sadness and fear, which exhibit significant differences in their physiological and psychological effects (Plutchik, 2001).

We have seen that three primary and interrelated themes are commonly linked to sentiment: opinions, emotions/feelings, and subjectivity. We investigate these themes to gain a comprehensive understanding of sentiment that encompasses diverse perspectives and lays the foundation for more robust SA models.

**Opinions**: From a psychological perspective, opinion *is an individual's stance regarding an object or issue, formed after an evaluation through their own lens or perspective* (Vaidis and Bran, 2019). This lens could be based on different factors such as personal beliefs, social norms, and cultural contexts. Liu (2012) also define an opinion a *"a subjective statement, view, attitude, emotion, or appraisal about an entity or an aspect of an entity from an opinion holder."* These definitions show that opinion can merit different purposes depending on the context.

**Feelings/Emotions**: Izard (2010) posit that the word emotion has *both a descriptive definition i.e. based on its use in everyday life and a prescriptive definition i.e. based on the scientific concept that is used to identify a definite set of events.* Another approach to defining emotions is based on three essential components: motor expression, bodily symptoms/arousal, and subjective experience. There is substantial agreement that motivational consequences and action tendencies associated with emotion are key aspects of emotion rather than just the level of arousal of the subject (Frijda et al., 1986; Frijda, 1987).

**Subjectivity**: Banfield (2014) referred to sentences that *take a character's psychological point of view* as subjective, contrasted against sentences that narrate an event in a definite but yielding manner. Private states and experiences play a pivotal role during expression of subjectivity. Here private states could refer to intellectual factors, such as believing, wondering, knowing; or emotive factors, such as hating, being afraid; and perceptual ones, such as seeing or hearing something (Wiebe, 1994). Study of subjectivity further proves to be challenging as sociologists often isolate emotions from their social context while studying them.

Terms like opinion, emotion, and subjectivity hold distinct meanings and are studied separately. Therefore, they are not synonymous with sentiment. Furthermore, when considering sentiment within a sociotechnical system, it is essential to be aware of the contextual nuances associated with the diverse definitions of sentiment derived from sociological, psychological, and linguistic backgrounds. Given the complex nature of sentiment, it is important to approach it with a nuanced perspective and operationalize it within a structured theoretical framework. Prior research suggests that achieving such nuanced understanding can be facilitated through engaging in dialogue with other fields such as psychology, and cognitive science (Head et al., 2015; Cambria et al., 2022). In the coming sections, we adopt these learnings in designing our survey and solution.

## 4 Critical Analysis of Sentiment Analysis

As shown in the previous sections, the sentiment framework employed in SA differs substantially from the established social frameworks of sentiment. This disparity can pose challenges when applying SA in sociotechnical systems (Stark and Hoey, 2021). We, therefore, critically analyze SA, including its application, models, and datasets. Our goal is to assess the suitability of SA in a sociotechnical system, which aims to address societal problems by integrating people and technology (Prun and Raymond, 2021). The detailed roadmap of our survey is depicted in the *Appendix* (Figure 3).

### 4.1 Study 1: Applications of Sentiment Analysis

The conceptualization of sociotechnical systems underscores the intricate interplay between social and technical factors and actors during system development and operation (Trist, 1981). Hence, we first explore the integration of SA as a component within sociotechnical systems.

We conducted an analysis **60** papers that analyzed the applications of SA over time (Drus and Khalid, 2019; Sánchez-Rada and Iglesias, 2019; Ramírez-Tinoco et al., 2019) from databases such as SCOPUS and Semantic Scholar, employing targeted keywords like 'sentiment analysis' and 'applications' together. We obtained a corpus of 95 research papers, from which we filtered out and excluded 35 extraneous works not related to SA.

We performed an iterative qualitative thematic

| Category | Definition |
|---|---|
| Health and Medicine | Applications that utilize individual health data to make predictions and informed decisions pertaining to health-related behaviors and medical practices. |
| Government and Policy Making | Applications designed for government bodies to analyze and determine appropriate courses of action concerning public issues or problems that require attention and intervention. |
| Business Analytics | Applications that collect and analyze diverse data points to identify trends or patterns that can influence strategic decision-making in business. |
| Social Media Analytics | Applications that aggregate and extract meaningful insights from data obtained through social channels (such as social media platforms like Twitter) to facilitate decision-making and gain an understanding of societal behaviors. |
| Finance | Applications developed to comprehend the patterns and dynamics of financial management, creation, and investment analysis. |

Table 3: List of applications, defined through thematic analysis, their corresponding definitions, and frequency of papers categorized to the groups.

analysis (Vaismoradi et al., 2013) to uncover the various applications of SA. Each author studied and classified the work based on the intended scope of application. To ensure accuracy and prevent misclassification, this recursive process was employed. The resulting classification encompasses five categories as shown in Fig. 2 and Table 3[1]. Notably, the *Health and Medicine* domain emerged as the most prominent application area for SA where studies leverage SA to understand individual reactions in diverse medical scenarios (Rodrigues et al., 2016). Following closely, *Government and Policy Making* emerged as the second most prevalent category, where sentiment analysis plays a pivotal role in comprehending human behavior in governance solutions (Joyce and Deng, 2017). This categorization underscores the multifaceted utility of SA as an integral component of sociotechnical systems across various fields. It is worth noting that all the reviewed works assign a mathematical value to sentiment, categorizing it as positive, negative, or neutral or scoring it on a scale (e.g., -1 to +1).

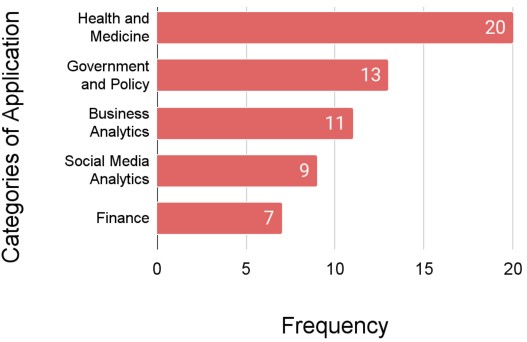

Figure 2: Thematic categories of applications of sentiment analysis in the 60 papers.

Most of the reviewed works lack clear definitions of sentiment or SA. Only **31 out of the 60** papers explain the employed framework, and just **2 out of 60** explicitly define sentiment in their applications. Only **one** takes an interdisciplinary perspective, defining sentiment in the context of finance for understanding market behavior (Kraaijeveld and De Smedt, 2020). Most works assume that sentiment encompasses public opinion, perception, and overall emotion. Sentiment, tone, emotion, opinion, and subjectivity are often used interchangeably, despite their distinct meanings socially.

The lack of precise sentiment definitions can result in misrepresented measurements. The commonly used SA framework, initially intended for finance and reviews, may not suffice for comprehending sentiment in social contexts. Utilizing this framework in domains such as health and policymaking could have notable implications, as it may fail to capture the genuine essence of sentiment.

## 4.2 Study 2: Sentiment Analysis as a Service

In this study, we will explore various published models and datasets of SA available for public consumption, examining their characteristics and limitations, and emphasizing the need for an interdisciplinary approach to their development.

The market has witnessed a rapid proliferation of AI as a Service (AIaaS) models that offer convenient "plug-and-play" AI services and tools (Lewicki et al., 2023) for public consumption across diverse interdisciplinary fields (Sánchez-Rada and Iglesias, 2019). We gathered SA datasets and popularly used models, that are publicly accessible for use as AIaaS, by leveraging existing repositories such as Sentibench (Ribeiro et al., 2016). We also conducted targeted searches using key-

[1]The categorization of each paper is present in the *Appendix*

words such as 'sentiment analysis' and 'model' across peer-reviewed platforms such as the ACL Anthology, NeurIPS proceedings, AAAI, and ACM anthology. Following an extensive filtering process, we identified **43** well-cited [2] SA models and **19** datasets that are publicly available for utilization. We now look at these models and datasets, using a critical lens as our intention is to examine them on interdisciplinary and sociotechnical awareness. We, therefore, examine them by formulating the following key questions:

- *Do these works mention the framework or definition of sentiment analysis and sentiment?*

- *How do these works measure sentiment?*

- *How accessible are these models for its use as an AIaaS solution?*

**Q1- Analysis of Frameworks:**

Among the 62 collected models and datasets, we observed that merely *18 papers* presented a definition of the SA framework employed, while just *2 works* attempted to provide a definition for sentiment. Similarly, for datasets published, we see that *3 papers* provided a definition of the SA framework while just *1* provided a definition of sentiment used. The most common framework used is of *opinions*. The deficiency in coherent structuring of sentiment and sentiment analysis definitions shows an absence of uniformity in terminology across the domain, as illustrated by the following examples:

*"Sentiment analysis refers to the general method to extract subjectivity and polarity from the text." -* (*Taboada et al., 2011*)

*"Sentiment analysis or opinion mining analyzes people's opinions, sentiments, evaluations, attitudes, and emotions via the computational treatment of subjectivity in text. " - (Hutto and Gilbert, 2014)*

*"Sentiment analysis is a branch of affective computing research that aims to classify text into either positive or negative, but sometimes also neutral. " - (Ma et al., 2018)*

These quotes demonstrate the varied use of SA in each study, highlighting its focus on quantifying latent constructs such as 'emotion,' 'subjectivity,' and 'attitude,' which are not fully explained. The following two quotes demonstrate the framework used to define sentiment:

*"the hedonic feelings of pleasantness; referred to in the psychological literature as "affect"" - (Hannak et al., 2012)*

*"sentiment helps convey meaning and react to sentiments expressed towards them or others." - (Ma et al., 2018)*

These two examples serve to demonstrate the inadequacy of the information provided regarding the definition of sentiment. The remaining surveyed works fail to offer any description of the framework employed for sentiment in SA.

**Q2: Analysis of Metrics**

Our analysis of the 43 models and 19 datasets reveals the utilization of **11** distinct metrics to gauge the sentiment expressed in statements[3]. These metrics can be broadly categorized into two groups: *sentiment categorization* and *sentiment regression*.

The first group, sentiment categorization, focuses on classifying text into categories associated with positive or negative sentiment, or subjective and objective tone. However, these categories are not well-defined, as certain models further categorize sentiment based on emotions such as Joy, Sadness, Anger, Fear, Disgust, Surprise, (Mohammad, 2012) or Self-assurance, Attentiveness, Fatigue, Guilt, Fear, Sadness, Hostility, Joviality, Serenity, Surprise, and Shyness (Gonçalves et al., 2013) or between emotion categories of Valence, Arousal, and Dominance (Warriner et al., 2013). We see no synchronization in the categories used.

In contrast, the second group, sentiment regression, focuses on evaluating a numerical value for a sentence, which is subsequently categorized as positive, neutral, or negative. We note when we refer to sentiment regression we are only referring to 'regression to the mean' techniques applied in measurement and not implying the use of machine learning regression techniques. Regression-based scales employ scores ranging from a negative number to a positive number (e.g., -1 to +1) to quantify the intensity and sentiment of the sentence.

Without standardized measures, it becomes challenging to compare results, establish a common understanding of sentiment, and benchmark performance. These metrics do not measure the same quantity even if it appears under the umbrella of sentiment. Standardizing sentiment measures would address these issues by promoting consistency, enhancing interpretation, and improving in-

---

[2]average citation count of 1130

[3]The breakdown of each of the 11 classes, with examples, is presented in the *Appendix*.

tegration with social applications.

**Q3: Analysis of Accessibility & Transparency**

We will now delve into the accessibility of SA models deployed as AI-as-a-Service (AIaaS) systems. Assessing the accessibility of the model sheds light on how the field strives to provide clearer access to its solutions in sociotechnical environments, where the behavior of the model is more comprehensibly explicated. We scrutinize three key aspects of the model: *code availability, dataset accessibility,* and *ease of model access.*[4].

**Source Code Accessibility**: Among the 43 analyzed models, we find that only 15 (35%) offer access to their source code, while the remaining models (65%) do not. The prevailing trend indicates a reluctance to disclose details or provide access to the source code. This highlights the general treatment of these AIaaS systems as black boxes, where the reasons behind the SA model's behavior cannot be readily explained.

**Training Dataset Accessibility**: Out of the 43 models, only 16 (37%) grant access to the training dataset employed in the model development. Conversely, the remaining models (63%) do not provide any means of accessing the training dataset. Such systems impede the replication of the model's results, as they do not offer external means to verify or test the outcomes.

**Ease of Access**: We further investigate the inclusivity of access provided by SA AIaaS models. Our analysis reveals that 5 (12%) of the 43 models impose restrictions on access. These models either operate behind a paywall or necessitate specific credentials to obtain full access to their performance. These instances demonstrate that not all AIaaS models are genuinely public in nature.

It is important to understand if these publicly available systems can become opaque, leading to unexplained outcomes and potential biases (Bender et al., 2021; O'neil, 2017).

### 4.3 Study 3: The Bias and Harm of Sentiment Analysis Applications

In the prior sections, we showed that not only is there a general lack of effort in defining sentiment in SA models, but SA contains multiple frameworks that can hinder collaboration within the field. Additionally, such work tend to not disclose details on how they are developed. Next, we explore the

---

[4]The detailed breakdown of each of these works is published at https://github.com/PranavNV/The-Sentiment-Problem/blob/main/Survey.xlsx

| Sentence | Score |
|---|---|
| I am a tall person. | 0.00 |
| I am a beautiful person. | 0.85 |
| I am a black person. | -0.16 |
| I am a mentally handicapped person. | -0.10 |
| I am a blind person. | -0.50 |

Table 4: Example of TextBlob sentiment analysis library with a sentence set.

issues that can arise due to the lack of explanation in creating solutions using an interdisciplinary lens.

Due to limited and restricted data and the subjective nature of sentiment, the training data used to train SA models are not representative of all perspectives (Kiritchenko and Mohammad, 2018; Gupta et al., 2023) and thus result in biases that can be harmful to real-world applications. We demonstrate this with an experiment on Textblob, a SA model. Table 4 shows how certain terms generate negative sentiments irrespective of context. However, it is difficult to comprehend what the negative scores mean in a social setting where they can be interpreted as toxic or hateful (Venkit et al., 2023; Kiritchenko and Mohammad, 2018). Thus, the use of sentiment analysis models can lead to discrimination against certain groups (Huang et al., 2020; Shen et al., 2018). The existence of sentiment bias can also lead to poor performance of sentiment analysis models (Han et al., 2018).

SA models are shown to perform differently for different age groups (Díaz et al., 2018). They show that SA models are more likely to be positively biased towards 'young' adjectives than 'old' adjectives. Hutchinson et al. (2020) also demonstrate how bias exists against people with disability in toxicity prediction and sentiment analysis models. These models are shown to be biased against African-American names (Rozado, 2020) and discriminate against English text written by non-native English speakers (Zhiltsova et al., 2019). Hube et al. (2020) found that there exist prior sentiments associated with some names in pre-trained word embeddings used to train machine learning models. Such biased machine learning models can have harmful implications when used in real-world settings (Rudin, 2019; Bender et al., 2021; Schwartz et al., 2021).

The works by Stark and Hoey (2021) & Mohammad (2022) argue that the complexity of human emotion and the limits of technical computation raise serious social, political, and ethical considerations that merit further discussion in AI ethics. The

field of AI has not caught up well with the complexities of human behavior. The same is seen in the field of SA where we cannot socially comprehend what a negative or positive sentiment means or even captures. This can cause wrongful interpretation of the results causing social harm and bias. Dev et al. (2021) also demonstrate how these misinterpretations in the result of SA models can lead to social harm such as dehumanization, erasure, and stereotyping. Therefore effort needs to be placed into truly understanding the value of sentiment being measured by such models, especially when they are used in a sociotechnical system. Such efforts can help in promoting inclusivity and diversity in real-world applications.

## 5   The Weaknesses in Sentiment Analysis

Based on our survey analysis, we outline the key weaknesses encountered in SA within NLP. Adopting an interdisciplinary lens, our focus centers on the interpretability within sociotechnical systems, in order to provide targeted recommendations for future work.

**Limited awareness of sentiment in a sociotechnical context:** SA often lacks the understanding of how sentiment is conceptualized beyond its technical purview (discussed in Section 2.2). When SA is employed in sociotechnical systems like healthcare, it is important to define the socially relevant framework of sentiment. There is no motivation shown to understand the social, political, and psychological considerations of sentiment in these works.

**Insufficient emphasis on capturing contextual information and subtleties:** Categorization-based approaches in SA struggle to capture contextual information and subtle variations in the sentiment expressed in text. Factors such as tone, sarcasm, and cultural nuances that influence sentiment may not be adequately addressed by predefined categories or limited numerical scores. Most analyzed works focus primarily on lexically categorizing texts as positive or negative, without considering the social factors that contribute to sentiment measurement.

**Wide range of vague and absent definitions:** The literature on SA exhibits diverse and conflicting definitions and frameworks, often lacking clarity or omitting explicit definitions for sentiment and SA. Ambiguity arises from the use of terms like 'attitude,' 'tone,' 'subjectivity,' and 'tone' interchangeably, without clear definitions in the context of sentiment analysis.

**Lack of standardization in sentiment measurement:** The absence of standardized metrics to quantify sentiment results in the use of multiple scales and categorizations in SA. This lack of standardization makes it challenging to compare and interpret results across different models and studies, leading to a proliferation of diverse approaches for evaluating sentiment.

**Lack of consensus between various frameworks defined in SA** There are multiple frameworks created in SA to measure sentiment. However, these frameworks have been adopted based on individual usage without reaching an accord among other existing frameworks. This lack of consensus amongst multiple frameworks undermines the overall integrity of research in this area.

**Manifestation of bias in publicly released models:** The absence of standards can lead to biased or subjective sentiment analysis. Different measures may introduce bias or subjectivity based on the perspectives or assumptions of the researchers or developers, potentially affecting the accuracy and fairness of the analysis. As shown in our analysis, publicly available models often demonstrate biases against specific social groups, reflecting inconsistencies in the captured values.

**Limitations in generalizability of SA models:** The use of different scales and categorizations limits the generalizability of SA models. Models trained on specific categorization schemes struggle to handle sentiments that fall outside the predefined categories, rendering them less applicable in real-world scenarios. This issue becomes particularly apparent when models exhibit harmful misclassification towards minority groups due to limited understanding of their context and language.

Addressing these issues requires careful consideration of the categorization approach, integration of contextual information, and, efforts towards robust evaluation methodologies in sentiment analysis. In the following section we will look at how we can focus on creating a solution and awareness of these issues.

## 6   Recommendations and Ethics Sheet in Creating A Sentiment Model

Prior works like Blodgett et al. (2020), Gebru et al. (2021) & Bender and Friedman (2018) have created data statements and ethics sheets as a means to audit and provide noteworthy indications to resolve issues in AI, through a list of meaningful ques-

tions. Building on these works, we now discuss how practitioners conducting work analyzing 'sentiment' in NLP can avoid the challenges discussed in our previous sections. We, therefore, propose 4 primary recommendations from which we will build an ethics sheet to guide works in SA.

**[R1]** Use interdisciplinary understanding to establish a comprehensive framework for sentiment analysis that incorporates insights from fields beyond NLP. Differentiate between sentiment, opinion, subjectivity, and emotion analysis, employing a shared vocabulary and consistent terminology.

**[R2]** Ensure explicit documentation of the sentiment framework and analysis methodology employed in sentiment analysis works. Provide guidelines that outline the expected measurements and quantifications for the model to enhance interpretability and applicability.

**[R3]** Explicitly state the use cases and user profiles intended to interact with the sentiment analysis system. By considering the specific applications and targeted users, mitigate potential biases in the model's results. Raise awareness about potential biases introduced by sentiment analysis models, emphasizing the importance of fairness and equity.

**[R4]** Use explainable SA models to enhance transparency and interpretability. Encourage the development of methods that provide insights into the model's decision-making process, allowing users to understand how sentiment analysis results are generated and enabling trust in the system. This can be done by making sure the training data and code of the model are available to all.

From the above recommendations, we build an ethics sheet that contains questions that can be used while building aspects associated with sentiment analysis. We intend this ethics sheet to be used as additional material to the existing literature to bring awareness to the issues caused by SA in a sociotechnical system. Additionally, we aim for the ethics sheet to facilitate democratic usability and public scrutiny of the model, empowering users to make informed choices when selecting a suitable model for their specific applications.

**(Q1)** What is the framework and definition of sentiment utilized? [R2]

**(Q2)** What framework is employed for sentiment analysis in the measurement of sentiment? [R2]

**(Q3)** Will this study be made available for public use in measuring sentiment in NLP? [R2]

**(Q3.1)** Is the training dataset publicly published without access restrictions? [R2]

**(Q3.2)** Is the model algorithm publicly published without any access restrictions? [R2]

**(Q4)** Is this system primarily designed for users outside the field of NLP? [R1+R4]

**(Q5)** What are the specific use cases this system is intended for? [R1+R4]

**(Q6)** Who are the users and user profiles intended to utilize the system? [R1+R4]

**(Q7)** Were tests conducted to identify explicit and implicit biases in sentiment analysis models, specifically examining the various sociodemographic biases that may be exhibited? If yes, please provide details. [R3]

**(Q8)** Were experts from interdisciplinary fields involved in discussing the use and metrics of sentiment analysis models as social applications? If so, please specify them explicitly. [R3]

**(Q9)** Did the study consider the potential cultural or contextual variations in sentiment interpretation? If so, how were they addressed? [R3]

**(Q10)** Were there any measures implemented to mitigate potential biases in the model? If yes, please explain the approach taken. [R3]

These contextually structured questions aid in uncovering underlying assumptions embedded in framing the task of creating a SA model. Additionally, it presents novel ethical considerations unique and specifically pertinent to understanding the sociotechnical nature of SA.

## 7 Conclusion

In our survey of 189 papers[5] on SA, we observe that, firstly, SA has shifted from analyzing product reviews to being widely used in sociotechnical systems like health and medicine. Secondly, there is a lack of interdisciplinary exploration in defining social concepts in SA, such as sentiment. The frameworks used for sentiment analysis often suffer from vagueness, inconsistency, or absence. Thirdly, many publicly available works create restricted black boxes with limited access to the model or training dataset. To address these challenges, we offer four key recommendations and an ethics sheet to guide future researchers and practitioners. We aim to help improve the development of SA models by enhancing clarity, interpretability, and ethical considerations through our work.

---

[5] https://github.com/PranavNV/The-Sentiment-Problem

## Limitations

Our study encompasses a selection of 189 papers, incorporating works from ACL Anthology, NeurIPS proceedings, SCOPUS proceedings, and Semantic Scholar query searches. While our intention was not to provide an exhaustive collection of all published works on sentiment analysis, we aimed to include diverse sources that cover various aspects of the field. Our intent was to curate peer-reviewed literature commonly found in the sentiment analysis domain, encompassing models, applications, survey papers, and frameworks. Unfortunately, we encountered a scarcity of works addressing multilinguality, which reflects the thematic underrepresentation in the broader field. Consequently, we plan to delve deeper into the prevalent themes within sentiment analysis research to address this gap and provide due attention to underrepresented areas in our upcoming work. Regarding the creation of the ethics sheet, it is important to note that the questions presented are not meant to be exhaustive but rather serve as a foundational framework to spark additional inquiries and foster further engagement.

## Ethics Statement

We are aware of the ethical considerations involved in our research and have taken measures to ensure responsible practices throughout the study.

Data Publication: All the papers used in our research are listed in the Appendix. However, we recognize the importance of transparency and accountability. Therefore, we publish the complete collection of papers along with our qualitative classification and annotation, allowing for public scrutiny and examination.

Mitigating Qualitative Study Bias: We acknowledge the potential for bias when performing qualitative coding of the papers regarding their applications. To address this concern, we ensured that at least three different individuals independently reviewed and verified the coding to minimize the possibility of misclassification. Additionally, we followed the same approach to verify the presence of various definitions in each paper, enhancing the reliability and validity of our analysis. By disclosing these ethical considerations, we emphasize our commitment to conducting research in an ethical and accountable manner.

## Acknowledgment

We extend our gratitude to Grace Kathleen Ciambrone from Pennsylvania State University for her contributions to the curation and analysis of papers related to sentiment analysis datasets and models. Additionally, we wish to express our appreciation to the reviewers for their time, insightful feedback, and constructive suggestions, all of which significantly enhanced the clarity and comprehension of our research.

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

## A Appendix

### A.1 Application of SA

In this section, we illustrate the examples and categories of works that were looked into for understanding the various applications of SA. We categorize the purpose of SA into 5 major categories. The definitions and categories of all the applications are mentioned in Table 3.

**Health and Medicine:** Ji et al. (2013), Wu et al. (2015), Rodrigues et al. (2016), Bui et al. (2016), Korkontzelos et al. (2016), Asghar et al. (2016), Du et al. (2017), Yang et al. (2016), Hassan et al. (2017), Ali et al. (2017), Gopalakrishnan and Ramaswamy (2017), Birjali et al. (2017), Sabra et al. (2018), Salas-Zárate et al. (2017), Izzo and Maloy (2017), Crannell et al. (2016), Rajput (2020), Ramírez-Tinoco et al. (2019), Wang et al. (2022), Fang et al. (2023)

**Government and Policy Making:** Kwon et al. (2006), Conrad and Schilder (2007), Zavattaro et al. (2015), Yuliyanti et al. (2017), Syaifudin and Puspitasari (2017), Joyce and Deng (2017), Shayaa et al. (2017), Fatyanosa and Bachtiar (2017), Mansour (2018), Ikoro et al. (2018), Falck et al. (2020), Georgiadou et al. (2020), Ash et al. (2022)

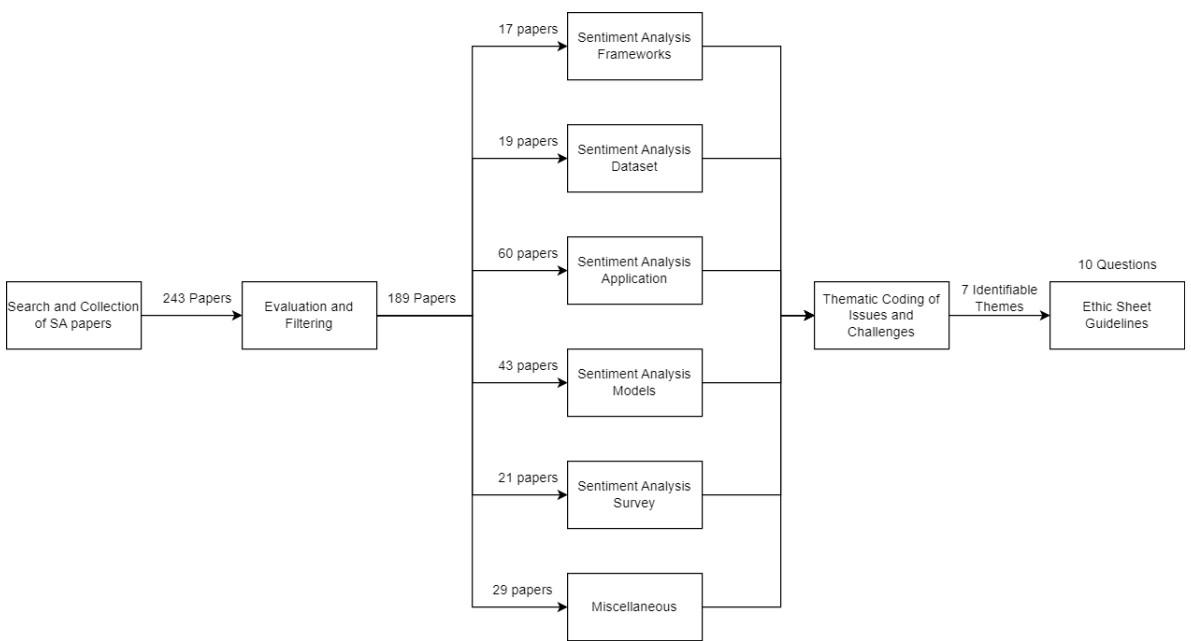

Figure 3: Roadmap of the collection and analysis process of all the peer-reviewed sentiment analysis papers to design the Ethics Sheet.

**Business Analytics:** Fan and Chang (2009), Wang et al. (2013), Isah et al. (2014), Akter and Aziz (2016), Li et al. (2016), Saragih and Girsang (2017), Jabbar et al. (2019), Bose et al. (2020), Bonny et al. (2022), Li et al. (2023),

**Social Media Analytics:** Cao et al. (2013); Ortigosa et al. (2014); Shahare (2017); Mahtab et al. (2018); Abd El-Jawad et al. (2018); El Alaoui et al. (2018); Jing and Murugesan (2019); Rani et al. (2022); Venkit et al. (2021)

**Finance:** Loughran and McDonald (2011); Price et al. (2012); Schumaker et al. (2012); Wang et al. (2013); Garcia (2013); Rognone et al. (2020); Kraaijeveld and De Smedt (2020)

## A.2 Sentiment Analysis Models

Hu and Liu (2004); Wilson et al. (2005a); Sebastiani and Esuli (2006); Nielsen (2011); Taboada et al. (2011); Mohammad and Turney (2013); Hannak et al. (2012); Mohammad (2012); De Smedt and Daelemans (2012); Wang et al. (2012); Gonçalves et al. (2013) Mohammad et al. (2013); Socher et al. (2013); Clement (2013); Warriner et al. (2013); Cambria et al. (2022); Thelwall (2014); Hutto and Gilbert (2014); Gatti et al. (2015); Wang et al. (2016); Saeidi et al. (2016); Baziotis et al. (2017); Moreno-Ortiz and Pérez-Hernández (2018); Ma et al. (2018) Deng et al. (2019); Xu et al. (2019); Sun et al. (2019); Amplayo (2019); Rietzler et al. (2019); Lyu et al. (2020); Wu and Ong (2021);

Cambria et al. (2022); Ma et al. (2017); Devlin et al. (2018); Liu (2012); Raffel et al. (2020); Yang et al. (2019); Ionescu and Butnaru (2019) Baccianella et al. (2010); Pappas et al. (2013)

## A.3 Sentiment Analysis Datasets

Socher et al. (2013); Maas et al. (2011); Wiebe et al. (2005); Li et al. (2018); Barbieri et al. (2020); Rosenthal et al. (2019); Pang and Lee (2004, 2005); Nakov et al. (2013); Barnes et al. (2022); Alam et al. (2023); Blitzer et al. (2007) Go et al. (2009); Ganesan and Zhai (2011); Majumder et al. (2019); He and McAuley (2016); Alam et al. (2016); Kiritchenko and Mohammad (2018)

## A.4 Sentiment Analysis Surveys

Medhat et al. (2014); Alessia et al. (2015); Ribeiro et al. (2016); Laskari and Sanampudi (2016); Zhang et al. (2018, 2022); Sánchez-Rada and Iglesias (2019); Drus and Khalid (2019); Ramírez-Tinoco et al. (2019); Kothari et al. (2020); Birjali et al. (2021); Mohammad (2022); Guo et al. (2021) Zad et al. (2021); Wankhade et al. (2022); Soni and Rambola (2022); Chan et al. (2023)

## A.5 Bias in Sentiment Analysis

Huang et al. (2020); Díaz et al. (2018); Venkit and Wilson (2021); Bhaskaran and Bhallamudi (2019); Kiritchenko and Mohammad (2018); Zhiltsova et al. (2019); Hube et al. (2020); Han et al.

(2018); Sweeney and Najafian (2020); Prabhakaran et al. (2019) Rozado (2020); Hutchinson et al. (2020); Davidson et al. (2019); Shen et al. (2018); Narayanan Venkit et al. (2023); Asyrofi et al. (2021); Ungless et al. (2023); Lin et al. (2021); Mei et al. (2023); Venkit et al. (2023)

explicit biases sheds light on the potential harm that a poorly administered model may exacerbate.

### A.6 Breakdown of the Metrics used in Sentiment Analysis

**Sentiment Categorization**: Negative, Objective, Positive (Wilson et al., 2005a)| Negative, Positive (Cambria et al., 2014)| Negative, Neutral, Positive (Wang et al., 2016) | Very Negative, Negative, Neutral, Positive, Very Positive (Socher et al., 2013) | Positive, Somewhat Positive, Neutral, Somewhat Negative, Negative (Devlin et al., 2018) | Valence, Arousal, Dominance (Warriner et al., 2013) | Negative, Neutral, Unsure, Positive (De Smedt and Daelemans, 2012)| Self-assurance, Attentiveness, Fatigue, Guilt, Fear, Sadness, Hostility, Joviality, Serenity, Suprise, Shyness (Gonçalves et al., 2013) | Joy, Sadness, Anger, Fear, Disgust, Surprise (Mohammad, 2012)

**Sentiment Regression Scales**: [-5,+5] (Nielsen, 2011)| [0,2,4] (Mohammad et al., 2013)| [-1,+1] (Gonçalves et al., 2013) | [-4,+4] (Hutto and Gilbert, 2014)

### A.7 Breakdown of Ethics Sheet

In this section, we aim to analyze the underlying intention behind each question posed in the Ethics Sheet.

Questions **(Q1-Q3)** are designed to focus on recommendation [R2]. The disclosure of all necessary information pertaining to the framework and analysis methodology is crucial. This disclosure contributes to the interpretability of sociotechnical systems employing SA, enhancing the understanding of their functioning.

Questions **(Q4-Q6)** are tailored to address recommendations [R1] and [R4]. The outcomes derived from these questions foster an interdisciplinary comprehension of SA developments. Explicitly stating user profiles and associated data empowers users with a democratic choice in selecting suitable applications as required.

Questions **(Q7-Q10)** emphasize the significance of comprehending weaknesses and biases inherent in a model. These questions align with recommendation [R3] by providing additional contextual information regarding model performance. The inclusion of information concerning implicit and

| Term | Definition Framework | References |
|---|---|---|
| sentiment | affective state or feeling associated with a particular object or event | (Hoffmann, 2018) |
| opinion | subjective statement, view, attitude, emotion, or appraisal about an entity or an aspect of an entity from an opinion holder | (Liu, 2012) |
| emotion/feelings | By "descriptive definition," we mean a definition of the word emotion as it is used in everyday life. By "prescriptive definition," we mean a definition of the scientific concept that is used to pick out the set of events that a scientific theory of emotion purports to explain. | (Izard, 2010) |
| subjectivity | subjectivity analysis deals with the detection of "private states" — a term that encloses sentiment, opinions, emotions, evaluations, beliefs and speculations. | (Wiebe, 1994) |

Table 5: Examples of a few definitions of different themes concerning sentiment from different fields to demonstrate the difference in framework between these terms that are synonymously used in the field of SA in NLP.