# OpenReview forum: "The Sentiment Problem: A Critical Survey towards Deconstructing Sentiment Analysis"
_EMNLP/2023/Conference — EMNLP 2023 Main_

### Official Review · Reviewer_8ryf · 2023-07-25

**Soundness:** 4

**Excitement:**

4: Strong: This paper deepens the understanding of some phenomenon or lowers the barriers to an existing research direction.

**Missing References:**

Measurement modelling framework references given above.

The authors may want to consider referencing some of the work on aspect-based sentiment analysis.

Paper on bias in proprietary sentiment analysis models against LGBTQ+ community - Ungless, E. L., Ross, B., & Belle, V. (2023). Potential Pitfalls With Automatic Sentiment Analysis: The Example of Queerphobic Bias.

**Paper Topic And Main Contributions:**

In this paper the authors highlight current issues with the way sentiment is conceptualised and measured in papers on “sentiment analysis” (SA). They demonstrate conflict both within the NLP literature, and between papers on SA and the conceptualisation of sentiment (and the related concepts of emotions, opinions and subjectivity) in the social science literature. They demonstrate that only half of papers applying SA, and less than half of papers proposing a model and dataset, provide a clear indication of which framework of SA they use, and only a handful actually define sentiment. The authors highlight issues of bias in existing SA models. They summarise the key weaknesses of SA - in addition to conflicting or missing definitions, there are the issues of subjective interpretation; lack of nuance amongst others. The authors then propose 4 recommendations and 10 related questions which can guide practitioners proposing new SA artefacts to be theoretically grounded and robust, and provide sufficient detail.

**Questions For The Authors:**

(A) In part 2 did you analyse 43 or 60 models because the figures given are inconsistent?
(B) To clarify, were all the models that you analysed published in peer-reviewed articles, including those without source code or made available only behind a paywall.
(C) Please clarify where the 189 papers mentioned in line 704 come from. Is this just all the papers you cite, papers found in your two systematic literature reviews, papers analysed across Studies 1-3?

**Reasons To Accept:**

This is a confidently written paper which highlights serious murkiness in the way sentiment analysis is currently conceptualised and conducted. The authors succinctly summarise the NLP and social science literature to demonstrate where lack of shared understanding, or even clear definitions of key concepts leads to issues related to validity, comparability and usefulness of measures.

The authors provide useful guidance for practitioners - I think the recommendations drawing attention to the impact of use cases, and cultural or contextual variations in sentiment interpretation, are particularly valuable for the community.

**Reasons To Reject:**

I found the definition of “framework” provided to be slightly too broad to be useful. For example, the distinction between providing a definition of sentiment, and defining the sentiment framework (which they say encompasses “a set of principles… for… defining a specific concept”), is unclear. I believe the distinction may be clearer if cast in the light of the measurement modelling framework which distinguishes conceptualisation and operationalisation of a concept such as sentiment (see Adcock and Collier, 2001; Jacobs and Wallach, 2021).

I also found the differences between the different technical frameworks for sentiment analysis difficult to distinguish, particularly “object’s orientation” vs “opinions or evaluation”, although the issue of conflicting definitions still comes across. I recommend the authors expand on why adopting one sentiment framework to a new domain “may not suffice” (L314), with concrete examples.

I found the limitations section to be superficial: it was more a statement of the fact the work is not exhaustive (what work is?) rather than an engagement with i.e. which areas the work fails to encompass, how the authors’ preconceptions may have limited the work etc

**Reproducibility:**

5: Could easily reproduce the results.

**Reviewer Confidence:**

4: Quite sure. I tried to check the important points carefully. It's unlikely, though conceivable, that I missed something that should affect my ratings.

**Typos Grammar Style And Presentation Improvements:**

L63 - missing parenthesis
L185 - commonly linked to sentiment in the technical literature (?)
L198 - a → as
L351 - Q1- → Q1:
L438 - superfluous punctuation
L568 - repetition of tone

---

> ### Author Rebuttal · Authors · 2023-08-29
>
> We extend our sincere gratitude for your feedback. We truly appreciate the time and effort you dedicated to this evaluation, and we are fully committed to addressing the points you have raised. Your recognition of the significance of works that shed light on the complexities within the field of sentiment analysis resonates with our own mission, and we are grateful for your acknowledgment of our efforts in this regard.
>
> To respond to your insights regarding the reviews, we wish to express our thanks for sharing references that illuminate the need for a clearer distinction between the concept of sentiment and the framework of sentiment analysis employed in NLP. The categorization of these aspects as conceptualization versus operationalization is a valuable perspective, and we are eager to incorporate this clarity into the introduction of our paper in the final draft.
>
> Regarding your second point, we wish to clarify that the ambiguity in distinguishing between various sentiment analysis frameworks simply reflects the existing ambiguity in the field. There is a need for future interdisciplinary research in order to develop standardized frameworks for sentiment analysis. Having said that, it is important to understand (as discussed in our work) that the diversified set of definitions that emerge from different fields leads to complexities in the standardization of frameworks. We attempt to underscore how existing frameworks often overlap and lack well-defined boundaries, resulting in a somewhat fluid landscape. These ad-hoc creations can pose challenges when applied beyond the confines of sentiment analysis. Your suggestion to include examples to emphasize this point is well-received, and we commit to incorporating such illustrations into the paper.
>
> In response to your specific queries:
>
> A. Study 2 consists of 43 papers focused on sentiment analysis models and 19 papers focused on sentiment analysis datasets, totaling 62 papers. We acknowledge the need to edit this detail within the section.
>
> B. Yes, all the papers we collected for our study were published in peer-reviewed journals or conference proceedings. We have included the comprehensive collection of these papers in the Appendix and supplementary material. Furthermore, we intend to make this collection publicly accessible alongside the final version of our paper.
>
> C. We appreciate your inquiry regarding the sources underpinning our analysis of the weaknesses in the field of sentiment analysis, as encapsulated in our paper and the three individual studies presented. To clarify, we conducted multiple literature reviews, each aligned with specific facets of sentiment analysis i.e., Sentiment Analysis Applications, Sentiment Analysis Models, Sentiment Analysis Datasets, Surveys and Meta-Analysis, and Frameworks (as shown in Table 1). These sources have been meticulously cited throughout our work and are reclassified in the Appendix for reference. We will ensure that this point is articulated more explicitly in our paper to enhance its clarity.
>
> Once again, we express our appreciation for your positive feedback and recognition of our work. Your insights as well as the additional references have been invaluable in shaping the refinement of our paper, and we look forward to incorporating these enhancements in our final version.

---

### Official Review · Reviewer_HzjA · 2023-07-27

**Soundness:** 4

**Excitement:**

4: Strong: This paper deepens the understanding of some phenomenon or lowers the barriers to an existing research direction.

**Paper Topic And Main Contributions:**

The paper describes work on surveying a large body of published sentiment analysis papers in literature to provide a sociotechnical perspective on recurring weaknesses and challenges present in the field. The author/s’ emphasize the motivation of the study by highlighting that the progress of sentiment analysis over the years cuts across beyond computing areas and into interdisciplinary sections such as finance, medicine, government, etc. One of the main challenges the survey highlights is the lack of concrete definitions and characterization of what a ‘sentiment’ is and how this differs from concepts such as opinion mining, emotion recognition, and semantic orientation, to name a few. The study is split into different analyses covering aspects of various frameworks assumed by studies for sentiment analysis, (technical/social) perceptions of sentiment tackled, mentioned applications of sentiment analysis across different fields, demonstration of existing biases in current sentiment analysis tools, and existing challenges still faced by these systems including lack of standardization, limited context, and generalizability. The study then proposes an ethics checklist that can be used by future sentiment analysis studies to possibly alleviate some of the mentioned challenges, such as lack of definition and mentions of target users of the system.

**Questions For The Authors:**

1. Like any survey paper, I think it’s worth mentioning the exact SCOPUS search string used for the study. This should also be included in the main sections of the paper. I tried using "TITLE-ABS-KEY ( "sentiment analysis" ) AND PUBYEAR > 2010” as mentioned in the Appendix, but this returns 29k hits. What additional parameters did the authors use to filter this? This is for SCOPUS, information on how documents were retrieved should also be added in the case of Semantic Scholar.

2. What are the authors’ inferences if terminologies used by research on sentiment analysis are often interchanged on paper but mean one particular framework (ex. polarity)? Is there harm in such a case?

3. What is the resolve for published works encompassing multiple applications by using multiple datasets in sentiment analysis? Do they count separately to different items in Figure 1?

**Reasons To Accept:**

The paper is well-written. I appreciate how readable the discussions are and how well-structured the flow of the narrative of the paper is, which makes it easier to evaluate and critique. Likewise, I do favor studies like these to be publicized and talked about in conferences as it gives a wider perspective and sanity check on the field of sentiment analysis for experts and beginner researchers alike. An ethics sheet, like the NLP checklist for most *ACL conferences that I personally find useful, will definitely help researchers ground the context and limitations of (future) studies using this, thus reducing vagueness and lack of transparency in applications of sentiment analysis.

**Reasons To Reject:**

I have no major issues to raise to possibly merit a rejection score for the paper. However, I do have some reservations and concerns regarding how the paper is angled. I advise the author/s to consider these points for the improvement of the wholeness of the idea and contribution of the study:

1. While reading the paper, I expected that the authors would thoroughly discuss language coverage and its related concepts (ex., code-switching, multilinguality, these are all qualified challenges in SA) and the scope of the critical survey regarding this factor. However, I’m quite disappointed that this was not mentioned at all. I don’t think this should be pushed back in the limitations section and claim that the survey is not exhaustive. This should be discussed outright in the collection phase and further expanded in the limitation section. Information on the language intricacies maybe even included as part of the checklist.

2. The notion of proposing an ethics checklist for a specific task implies to the reader that other tasks in NLP may also merit the construction of such a list as they all face different challenges and risks when used across disciplines such as education, government, etc. As an NLP/SA researcher, I would understand perfectly why this resource is needed, however, for beginners and non-technical readers, they might not appreciate the notion of a checklist outright if this is only done in SA (excluding checklists for general AI experiments and datasets). Thus, as part of the motivation of the study, discussing that NLP tasks are doing sanity checks via an ethics checklist is important. I believe this would give more context and would further emphasize the need for an ethics checklist for sentiment analysis.

3. I believe the study needs a good Figure 1. Upon reading the technical and social perception of sentiment analysis, confusion may arise with the terminologies (and their overlapping meaning) and their relation to one another. Readers may require some visual aid to connect the dots and terminologies to entertain thoughts such as “If I have an opinion on something, then that means I may have strong feelings or emotions about this something. I can express this feeling through texts with emoticons” So this does not necessarily mean that opinions and emotions are distinct concepts in SA, as the authors may have framed the paper with definitions from multiple works.

**Reproducibility:**

4: Could mostly reproduce the results, but there may be some variation because of sample variance or minor variations in their interpretation of the protocol or method.

**Reviewer Confidence:**

4: Quite sure. I tried to check the important points carefully. It's unlikely, though conceivable, that I missed something that should affect my ratings.

---

> ### Author Rebuttal · Authors · 2023-08-29
>
> We thank you for taking the time to review our work and for sharing your valuable insights. We appreciate and mirror your commitment to championing the publication of such works in conferences like EMNLP, as well as your recognition of the necessity for rigorous scrutiny in the NLP domain, particularly considering its growing societal implications.
>
> Your critique concerning the inclusion of multilingual aspects in sentiment analysis is well-founded. Our original intent was to curate peer-reviewed literature commonly found in the sentiment analysis domain, encompassing models, applications, survey papers, and frameworks. Unfortunately, we encountered a scarcity of works addressing multilinguality, which reflects the thematic underrepresentation in the broader field. Consequently, we plan to delve deeper into the prevalent themes within sentiment analysis research to address this gap and provide due attention to underrepresented areas. Your suggestion to integrate multilinguality into our checklist aligns with our objectives, and we commit to implementing these changes.
>
> Regarding your second concern regarding introducing the concepts of the ethics sheets to beginners, we appreciate the feedback. While we did mention and introduce various works on ethics sheets in the NLP domain at the opening of the "Recommendations and Ethics Sheet in Creating a Sentiment Model" section, we will further emphasize this in our introduction and motivation.
>
> We understand the need to incorporate a figure illustrating the various frameworks of sentiment analysis. The decision not to include it in the current 8-page draft was driven by the need to allocate space to address other pertinent points. Rest assured, we will integrate this figure into the final 9-page version of the paper.
>
> In response to your specific questions to the authors:
>
> 1. We appreciate your inquiry regarding our search methodology. To clarify, our search did not exclusively focus on sentiment analysis alone but rather encompassed corresponding fields of published works in sentiment analysis, i.e., its applications, models, frameworks, and survey papers. Our selection criteria involved ensuring that the collected papers were published in peer-reviewed journals and conferences, followed by a manual evaluation if they fit into our categories. Each category contained its unique search inquiry and data source, which has been mentioned at the outset of each section. We have developed a comprehensive flowchart outlining our collection process for each of these categories, which we will include in the paper's Appendix. Additionally, we will make this process more explicit in our paper.
>
> 2. Your concern about the interchangeability of sentiment-related terms in different research fields is well taken. We believe that this ambiguity can introduce confusion and then harm, especially given the diverse interpretations of these terms in various contexts. It is imperative to address this issue, as it can lead to misinterpretations of models and results. When applications of sentiment analysis are used in social platforms with social actors/entities, terms such as opinion, polarity, and subjectivity can be interpreted differently, leading to the misevaluation of human behaviors. This can lead to wrongful inferences of the population due to the disciplinary divide of shared vocabulary. We emphasize this point in our paper, particularly in the "Weaknesses in Sentiment Analysis" section. Additionally, we emphasize in Section 2 that sentiment-related terms do not have a common definition in the field of sentiment analysis. There is, therefore, a need for interdisciplinary research to develop standardized frameworks.
>
> 3. We acknowledge your observation regarding the potential presence of works that overlap applications across multiple fields. We did not encounter such papers in our literature sample of sentiment analysis application papers. Our focus was on papers that tackled sentiment analysis by the type of work they are situated in and the theme of works that are commonly published in the field. While there are indeed papers that develop sentiment analysis models for general use, they were categorized as model papers in our collection. The application-themed papers were works that used sentiment analysis as a solution for service in that particular field. We will ensure this distinction is clarified in our paper. You can see the categorization of these works in our supplementary material as well. This will be published publicly, along with the paper, for access to our literature collection.
>
> Once again, we thank you for your positive input and encouragement for our work in this conference.

---

### Official Review · Reviewer_tS1k · 2023-08-05

**Soundness:** 5

**Excitement:**

4: Strong: This paper deepens the understanding of some phenomenon or lowers the barriers to an existing research direction.

**Paper Topic And Main Contributions:**

This position paper explores the social impact of sentiment analysis, one of the most prevalent NLP applications in the world (perhaps before ChatGPT). The study conducts a critical survey and categorization of existing "Sentiment Analysis" research and develops an Ethics Sheet for sentiment analysis to promote a safer and more transparent development of sentiment analysis systems.


**Questions For The Authors:**

It would be nice to provide a sample-filled example of the ethics sheet for a sentiment analysis study, like the "Show Your Work" paper [1]? This will help people understand how to implement your ethics sheet in practive.

[1] [Show Your Work: Improved Reporting of Experimental Results](https://aclanthology.org/D19-1224) (Dodge et al., EMNLP-IJCNLP 2019)

**Reasons To Accept:**

- Overall, this study's suggestions and ethics sheet will (or should) have a long-lasting impact. The study provides important insights to the community.


**Reasons To Reject:**

- I'm not 100% sure if this study is a good fit for the "empirical method" of NLP conferences, but this is just nitpicking.

**Reproducibility:**

N/A: Doesn't apply, since the paper does not include empirical results.

**Reviewer Confidence:**

4: Quite sure. I tried to check the important points carefully. It's unlikely, though conceivable, that I missed something that should affect my ratings.

---

> ### Author Rebuttal · Authors · 2023-08-29
>
> Thank you for your time and invaluable insights in evaluating our paper! We value and agree with your viewpoint on our survey's importance in promoting safer and more transparent development of sentiment analysis applications. It is important to have such critical analysis done on the field to demonstrate how the same can be improved for a safer and less biased environment. Your engagement with our work is greatly appreciated.
>
> Answer to your questions:
>
> We extend our thanks for raising the pertinent concern regarding the inclusion of a comprehensive illustration of the ethics sheet. This is a very relevant question, and consequently, in our upcoming work, we plan to conduct an in-depth analysis of these cases using established sentiment analysis models, thereby furnishing a platform to exhibit our empirical findings. We acknowledge your suggestion and will incorporate a sample of the ethics sheet use case in our Appendix. Thank you for sharing a relevant reference in this context, which will enrich the quality and context of our work.

---

### Meta-Review · Area_Chair_YJeK · 2023-09-18

**Recommendation:** 5

**Metareview:**

This work provides a survey of papers on sentiment analysis and provides a sociotechnical perspective on the area, discussing the issues and challenges. These challenges include the lack of a concrete definition of “sentiment analysis” used within NLP and across NLP and social science, challenges in existing sentiment analysis models, such as biases and limited generalizability among others. Lastly, it provides four pieces of advice and 10 related questions to mitigate the challenges it discusses.

Reviewers appreciated that this work provides a succinct summary of the task of sentiment analysis, a careful discussion of its issues, and a practical guideline for (at least partially) mitigating the issues. The presentation is also clear, which is especially important for this type of paper.

The reviewers do not seem to have major concerns but noted that a main figure presenting an overview of the survey would be helpful, some terms like “framework” can be better clarified, and the limitations section can provide a deeper discussion, among others.

---

### Decision · Program_Chairs · 2023-10-07

**Decision:**

Accept-Main

**Comment:**

This work provides a survey of papers on sentiment analysis and provides a sociotechnical perspective on the area, discussing the issues and challenges. These challenges include the lack of a concrete definition of “sentiment analysis” used within NLP and across NLP and social science, challenges in existing sentiment analysis models, such as biases and limited generalizability among others. Lastly, it provides four pieces of advice and 10 related questions to mitigate the challenges it discusses.

Reviewers appreciated that this work provides a succinct summary of the task of sentiment analysis, a careful discussion of its issues, and a practical guideline for (at least partially) mitigating the issues. The presentation is also clear, which is especially important for this type of paper.

The reviewers do not seem to have major concerns but noted that a main figure presenting an overview of the survey would be helpful, some terms like “framework” can be better clarified, and the limitations section can provide a deeper discussion, among others.